# Involvement of Bcl-2 Family Proteins in Tetraploidization-Related Senescence

**DOI:** 10.3390/ijms24076374

**Published:** 2023-03-28

**Authors:** Daniel Barriuso, Lucia Alvarez-Frutos, Lucia Gonzalez-Gutierrez, Omar Motiño, Guido Kroemer, Roberto Palacios-Ramirez, Laura Senovilla

**Affiliations:** 1Unidad de Excelencia Instituto de Biología y Genética Molecular (IBGM), Universidad de Valladolid—CSIC, 47003 Valladolid, Spain; 2Centre de Recherche des Cordeliers, Equipe Labellisée par la Ligue Contre le Cancer, Inserm U1138, Institut Universitaire de France, Université Paris Cité, Sorbonne Université, 75006 Paris, France; 3Metabolomics and Cell Biology Platforms, Institut Gustave Roussy, 94805 Villejuif, France; 4Institut du Cancer Paris CARPEM, Department of Biology, Hôpital Européen Georges Pompidou, AP-HP, 75015 Paris, France

**Keywords:** Bcl-2 family proteins, senescence, apoptosis, tetraploidy, cancer

## Abstract

The B-cell lymphoma 2 (Bcl-2) family of proteins is the main regulator of apoptosis. However, multiple emerging evidence has revealed that Bcl-2 family proteins are also involved in cellular senescence. On the one hand, the different expression of these proteins determines the entry into senescence. On the other hand, entry into senescence modulates the expression of these proteins, generally conferring resistance to apoptosis. With some exceptions, senescent cells are characterized by the upregulation of antiapoptotic proteins and downregulation of proapoptotic proteins. Under physiological conditions, freshly formed tetraploid cells die by apoptosis due to the tetraploidy checkpoint. However, suppression of Bcl-2 associated x protein (Bax), as well as overexpression of Bcl-2, favors the appearance and survival of tetraploid cells. Furthermore, it is noteworthy that our laboratory has shown that the joint absence of Bax and Bcl-2 antagonist/killer (Bak) favors the entry into senescence of tetraploid cells. Certain microtubule inhibitory chemotherapies, such as taxanes and vinca alkaloids, induce the generation of tetraploid cells. Moreover, the combined use of inhibitors of antiapoptotic proteins of the Bcl-2 family with microtubule inhibitors increases their efficacy. In this review, we aim to shed light on the involvement of the Bcl-2 family of proteins in the senescence program activated after tetraploidization and the possibility of using this knowledge to create a new therapeutic strategy targeting cancer cells.

## 1. Introduction

Members of the B-cell lymphoma 2 (Bcl-2) family proteins are known to be the main regulators of the intrinsic apoptosis pathway. Antiapoptotic proteins prevent the triggering of apoptosis, whereas proapoptotic proteins favor the induction of programmed cell death. Therefore, the presence or absence of these proteins is determinant in the survival and resistance of cells to an apoptotic signal. However, Bcl-2 family proteins also exert other noncanonical functions affecting cellular senescence, bioenergetic metabolism and redox homeostasis [1]. In this review, we will focus on the role of Bcl-2 family proteins in senescence [2]. Cellular senescence, hereafter referred to as “senescence”, is a prolonged cell cycle-induced arrest of previously replicative cells. Senescence may be an alternative way to apoptosis for suppressing cell growth. Both apoptosis and senescence play a key role in cancer. While apoptosis eliminates cancer cells, senescence arrests cancer cells in a permanent state of non-division.

Tetraploidy is considered a precancerous stage. In 2010, Davoli et al. proposed that, at the onset of tumorigenesis, cells may undergo permanent telomere dysfunction accompanied by DNA damage that would result in permanent cell cycle arrest. Cells that manage to escape this arrest, due to the absence of p53, would undergo endoreplication, giving rise to tetraploid cells [3]. Additionally, chemotherapeutic drugs include antimitotic agents such as taxanes and vinca alkaloids, which mechanism of action is based on tubulin binding. The prolonged use of antimitotic agents can cause chronic arrest in mitosis, cell death or mitotic slippage, leading to tetraploidy, apoptosis and/or senescence [4,5,6]. These chemically induced tetraploid cells, with functional p53, arrest their cell cycle G1, known as “tetraploidy arrest”, and undergo senescence [7]. Antimitotic agents are used in anticancer chemotherapy and can induce the tetraploidization of malignant cells in vivo [8] and promote therapy-induced senescence (TIS). Senescent cells originating from TIS can generate tumor resistance or lead to cancer recurrence if the stability of the proliferative arrest is weakened [9]. Targeting these senescent cells before they escape from proliferative arrest is key to improving the success of cancer therapies.

Bcl-2 family proteins are key in the resistance to chemotherapeutic agents in some cancer cells [10], regulate cancer cell survival after prolonged mitotic arrest [11] and are involved in the persistence of tetraploid cells by preventing the senescence program associated with tetraploidization [12]. Therefore, a possible approach is the use of inhibitors of the Bcl-2 family that fall into the class of “senolytics”, i.e., compounds endowed with the capacity to kill senescent cells [13].

## 2. Apoptosis and Senescence: Two Ways to Suppress Cell Proliferation

While apoptosis is important for morphogenetic processes that take place during embryonic development [14], senescence occurs during the normal aging process and is transiently induced during tissue development and remodeling [15]. In addition to their role in physiological conditions, apoptosis and senescence also play a role in pathophysiological situations, such as cancer. On the one hand, the suppression of cell death by apoptosis can cause cancer. On the other hand, most chemotherapeutic drugs induce cell death by apoptosis [16]. In turn, senescence can be induced by a variety of cellular stresses, including some chemotherapeutics, as a tumor suppressor mechanism [15].

Senescence is different from other forms of cell cycle arrest, such as quiescence, terminal differentiation or exhaustion. Senescent cells are characterized by (1) a long-lasting growth arrest, by (2) the expression of antiproliferative molecules and (3) the activation of damage-sensing signaling pathways [17]. Senescence is triggered by DNA damage response or stress signaling. Initially, a p53/p21^Waf1/Cip1^-mediated cycle arrest occurs. The persistence of stressful stimuli over time results in the decline of p53/p21^Waf1/Cip1^, as opposed to an increase in the p16^INK4a^/retinoblastoma protein (pRb) [18]. Other markers of senescent cells are increased lysosomal β-galactosidase activity and the secretion of proinflammatory cytokines, termed the senescence-associated secretory phenotype (SASP). SASP production is regulated by the mammalian target of rapamycin (mTOR) and is dependent on the nuclear factor kappa-light-chain-enhancer of activated B cells (NF-κB) and p38 mitogen-activated protein kinase (p38MAPK) [17,19,20]. The different subfamilies of Bcl-2 proteins play a role in senescence, generally distinct from that in apoptosis, and are detailed below (Figure 1). This includes the antiapoptotic proteins (Bcl-2, B-cell lymphoma extra-large (Bcl-xL), Bcl-2-like protein 2 (Bcl-w) and Induced myeloid leukemia cell differentiation protein (Mcl-1)) and the proapoptotic proteins, which can be divided into two groups: the multidomain proteins (Bcl-2 associated x protein (Bax), Bcl-2 antagonist/killer (Bak) and Bcl-2 related ovarian killer (Bok)) and the BH3-only proteins (such as Bcl-2 interactin mediator of cell death (Bim), Bcl-2 associated agonist of cell death (Bad), Phorbol-12-myristate-13-acetate-induced protein 1 (Noxa), p53 upregulated modulator of apoptosis (Puma), BH3 interacting-domain death agonist (Bid), Bcl-2-interacting killer (Bik), Bcl-2-modifying factor (Bmf), Bcl2/adenovirus E1B 19 kDa protein-interacting protein 3 (Bnip3) and Activator of apoptosis hara-kiri (Hrk)).

## 3. Role of the Bcl-2 Family in Senescence

The relationship between Bcl-2 family proteins and senescence is dual. On the one hand, the expression of these proteins affects the entry into senescence. On the other hand, the entry into senescence modulates the expression of these proteins. Due to modifications in the expression of Bcl-2 family proteins, senescent cells are more resistant to apoptosis [2]. Most of the literature focuses on the involvement of the antiapoptotic protein subfamily in senescence. However, there are also works that focused on proapoptotic protein subfamilies or even studying the Bcl-2 family proteins globally [2]. Several studies have described the relationship of the Bcl-2 family of proteins to induced senescence, especially senescence induced by chemotherapeutic agents, and physiological aging [19]. Therefore, targeting senescent cells seems to be an important approach to eliminate these malignances. 

Cellular models have recently allowed the study of alterations in the expression of Bcl-2 family proteins in relation to senescence in different cell lines and in response to a variety of stimuli. For instance, the conditionally immortalized proximal tubule epithelial cell line overexpressing organic anion transporter 1 (ciPTEC-OAT1) shows a remarkable upregulation of SASP factors and p21^Waf1/Cip1^, as well as upregulation of Bcl-2, Bid and Bax and downregulation of Mcl-1, Bad, Bak and Bim, when entering senescence at 37 °C [21]. Senescence initiated by the suppression of heat shock protein 70-2 in epithelial ovarian cancer cells is characterized by an increased expression of p21^Waf1/Cip1^ and p16^INK4a^; upregulation of Bax, Bim, Bak, Bad, Bid, Puma and Noxa and downregulation of Bcl-2, Bcl-xL and Mcl-1 [22]. Doxorubicin (Dox) treatment reduces cell viability and increases the number of senescent cancer cells, which show elevated levels of p16^INK4a^, enhanced expression of Bcl-xL and Bim and reduced expression of Bax, Bak, Bid and Puma [23].

### 3.1. Antiapoptotic Bcl-2 Proteins

Accumulated evidence has demonstrated that antiapoptotic Bcl-2 proteins are critical for senescence establishment [1]. While high levels of Bcl-2 and Bcl-w can trigger senescence in response to several stimuli, such as DNA damage or hypoxia, high levels of Bcl-xL and Mcl-1 may prevent the cells from entering into oncogene-induced senescence (OIS) and TIS. Senescent cells, however, present high levels of Bcl-2, Bcl-xL and Mcl-1 that halt the ability of the cells to follow a programmed cell death. For instance, expression levels of Bcl-w, Bcl-xL and Bcl-2 are increased in DNA damage-induced senescence by etoposide and ionizing irradiation [24], which are known to increase the presence of tetraploid cells [25,26]. In turn, subsequent studies have shown that the presence of Bcl-2, Bcl-w and Bcl-xL underlies the resistance of senescent cells to apoptosis [24,27]. In both cancer and senescence, overexpression of Bcl-2 counteracts the proapoptotic genes Puma and Noxa, thereby limiting apoptosis [28]. However, Baar et al. showed reduced Bcl-2 but upregulated Puma and Bim in genomically stable primary human lung myofibroblasts IMR90 induced to undergo senescence by ionizing radiation, suggesting that IMR90 senescent cells are destined for death by apoptosis, but somehow, the execution of the death program is impaired. In this case, senescent IMR90s showed increased forkhead box protein O4 (Foxo4), which can regulate p21^Waf1/Cip1^ expression in senescent cells. Through p21^Waf1/Cip1^, p53 can induce p16^Ink4a^–independent cell cycle arrest in senescent IMR90s [29].

Entry into, and the maintenance of cells in, senescence depends on the upregulation of Bcl-2, meaning that senescence is associated with elevated levels of Bcl-2 [24,30]. Moreover, the overexpression of Bcl-2 potentiates senescence cancer cells, such as in K562 leukemia cells [31]. Therefore, Bcl-2, which can inhibit both apoptosis triggering and proliferation, causes a senescence-like phenotype [32]. The overexpression of Bcl-2 correlates with cell cycle arrest, which could promote senescence [33]. In fact, cell cycle arrest in G1 is mediated not only by the overexpression of Bcl-2 but also by the inhibition of cyclin-dependent kinase (Cdk)2 activity and induction of p27^Kip1^ [34]. Indeed, the overexpression of Bcl-2 upregulates the levels of p27 ^Kip1^ and the nucleolar phosphoprotein p130, a member of the pRb pocket protein family, which forms repressive complexes with the transcription factor E2F4, inhibiting its release and thus preventing cell cycle progression in quiescent fibroblasts [35]. Several types of stimuli require the presence of Bcl-2 to induce senescence, such as (1) DNA damage and serum starvation, through p38MAPK, in OI [36]; (2) hypoxia-induced senescence, independent of p53 and p16^INK4a^ [37]; and (3) chemotherapy-induced cell growth inhibition that involves the accumulation of p53/p16^INK4a^ and senescence markers [38]. Lee et al. first described in 2010 how inhibition of the c-Jun N-terminal kinase (JNK), a regulator of oxidative DNA damage, by SP600125 induces premature senescence. The inhibition of JNK results in the dephosphorylation of Bcl-2, followed by the accumulation of ROS. The increased production of ROS induces the DNA damage response (DDR), leading to cell cycle arrest. The inhibition of cell cycle progression induced by SP600125 treatment is characterized by the upregulation of p53 and p21^Waf1/Cip1^ and downregulation of pRb, as well as an increase in the inactive phospho-cell division control 25C (P-Cdc25C) phosphatase and a decrease in the cyclin B and Cdk2 levels [39]. Of note, SP600125 induces G2/M cycle arrest and an increase in aneuploid cells [40,41] (Figure 2A). Conversely to Bcl-2 or Bcl-xL, little is known about the role of Bcl-w in senescence. However, it is known that Bcl-w overexpression enhances cellular senescence by activating the p53/neurogenic locus notch homolog protein 2 (Notch2)/p21^Waf1/Cip1^ axis [42].

Interestingly, Bcl-xL has a dual effect on senescence. The natural upregulation of Bcl-xL during megakaryocyte differentiation or genetically overexpressed Bcl-xL in MEFs and in primary cultures of human lymphocytes reduce entry into senescence [43]. However, it has been shown that several cancer senescent cell types, such as triple-negative breast cancer cell lines and pilocytic astrocytoma tumor cells, exhibit high levels of the Bcl-xL protein [43,44], as well as in both pancreatic intraepithelial neoplasia (PanIn) and pancreatic ductal adenocarcinoma (PDAC) [45]. The presence, or induced upregulation, of Bcl-xL reduces entry into senescence stimulated by various stimuli. For instance, (1) the overexpression of Bcl-xL suppresses OIS in low-grade PanIn and apoptosis in high-grade PanIn [45]; (2) Bcl-xL blocks p38MAPK activation and inhibits senescence induction by preventing p53-induced ROS generation [46]; (3) the induction of DNA damage causes cell cycle arrest at the G2/M checkpoint, as well as translocation of Bcl-xL to the nucleus occurs, where it binds to Cdk1, inhibiting its kinase activity and stabilizing the senescence program [47]; (4) treatment with the topoisomerase I inhibitor SN38 induces and maintains stable p53- and p21^Waf1/Cip1^-dependent growth arrest due to increased Bcl-xL expression [48]; and (5) CCC-021-TPP, the novel pyrrole–imidazole polyamide targeting a specific mutation in mitochondrial DNA, causes cellular senescence accompanied by significant induction of the antiapoptotic Bcl-xL [49]. Noteworthy, an increased Bcl-xL expression contributes to the protection against apoptosis in the human colon cancer cell line HCT116 [48]. In turn, the ablation of Bcl-xL decreased the survival of radiated glioblastoma multiforme (GBM) cells [50], as well as induced OIS and apoptosis in PDAC [45]. Moreover, permanent cell cycle arrest in response to OIS generally occurs through the combined activation of the p53/p21^Waf1/Cip1^ and p16^INK4a^-pRb pathways. Malignant cells having escaped OIS rely on survival pathways induced by Bcl-xL/Mcl-1 signaling [51]. Therefore, not only the overexpression of Bcl-xL represses the entry into senescence but senescent cells also show elevated levels of Bcl-xL preventing apoptosis [43] (Figure 2B).

As Bcl-xL, Mcl-1 acts as a senescence inhibitor, since the overexpression of Mcl-1 in tumor cells is crucial for blocking the induction of senescence [52]. Recently, Troiani et al. showed that senescent tumor cells depend on Mcl-1 for their survival. Interestingly, Mcl-1 is upregulated in senescent tumor cells, including those expressing low levels of Bcl-2 [53]. In a mechanism such as that described for Bcl-xL, treatment with G2/M blocking agents increases the interaction between a shortened form of the Mcl-1 polypeptide, mainly located in the nucleus, with Cdk1, reducing its kinase activity and inhibiting cell growth [54] (Figure 2B). During extended mitotic arrest, Mcl-1 has been identified as a critical factor to determine whether cells trigger apoptosis or mitotic slippage [55]. The overexpression of Mcl-1 inhibits TIS and promotes tumor growth, whereas the downregulation of Mcl-1 delays tumor growth in vivo [52]. The anti-TIS function of Mcl-1 can be inhibited by a loop domain mimetic peptide [56]. Mcl-1-regulated TIS depends on the generation of ROS—more specifically, mitochondrial ROS—and subsequent activation of the DNA damage response. Mcl-1 prevents the expression of NADPH oxidase 4, limiting its availability in mitochondria and thus decreasing mitochondrial ROS production during TIS [57].

### 3.2. Multidomain Proapoptotic Bcl-2 Proteins

Aging is associated with the balance between Bax and Bcl-2 expression. However, this balance seems to be different, depending on the cell type or the organism. In mice prone to accelerated senescence (SAMP8 mice), decreased Bcl-2 expression and increased Bax expression are observed [58]. In turn, senescent human diploid fibroblasts show high levels of Bcl-2 and low expression of Bax, which is associated with resistance to oxidative stress-induced apoptosis [59]. Thus, Bax is the most studied multidomain proapoptotic protein in senescence. Bax upregulation has been observed under different circumstances in DNA damage-induced senescence. For instance, the knockdown of Cdk2-associated protein-1 (Cdk2ap1) increases the percentage of cells exhibiting DNA damage characterized by γ-H2AX, as well as increased p53/p21^Waf1/Cip1^ and Bax, which reduces proliferation and induces premature senescence in primary human dermal fibroblasts [60]. The combined treatment of AMG 232, a potent small molecule inhibitor that blocks the interaction of mouse double minute 2 homolog (Mdm2) and p53, and radiation results in the accumulation of γ-H2AX-related DNA damage, a significant increase in Bax expression and induction of senescence in human tumors [61]. The treatment of human breast cancer MCF-7 cells with metformin or phenformin induces increases in p53 protein levels and p21^Waf1/Cip1^ and Bax transcription in a dose-dependent manner, leading to senescence [62]. Pancreatic cancer is associated with the elevated expression of cyclin B1 and Mdm2, as well as lower expression of Bax and p21^Waf1/Cip1^. However, the silencing of cyclin B1 decreases proliferation and the proportion of cells in the S phase while increasing apoptosis, senescence and the proportion of cells in the G0/G1 phase. This increase in senescence was accompanied by enhanced levels of p21^Waf1/Cip1^ and Bax [63]. Overexpression of the inhibitor of growth protein 5 (Ing5) causes (1) the suppression of proliferation and induction of G2/M arrest, (2) apoptosis, (3) senescence and (4) chemoresistance to cisplatin and paclitaxel in human primary GB cell line U87. At the molecular level, overexpression of Ing5 in the U87 line results in a lower expression of Cdc2 and Cdk4 but higher expression of p21^Waf1/Cip1^, p53 and Bax [64] (Figure 3A).

Additionally, the absence of Bax and Bak can result in the failure of mitotic cell death or a delay in cell division. In both situations, it should be followed by cell cycle arrest and senescence in the tetraploid G1 phase. In addition, Bim and Noxa are involved in the activation of Bax and Bak in mitotically arrested cells [65].

### 3.3. BH3-Only Proapoptotic Bcl-2 Proteins

In general, the involvement of proapoptotic BH3-only proteins is unclear and may even be controversial in senescence. As described below, depending on the circumstances, the expression of these proteins may be increased or reduced depending on the context. Therefore, the implication of these proteins in senescence does not seem to be decisive and could be merely an accessory role. 

MEFs expressing p53R1752P, a hypomorphic mutation that favors senescence versus apoptosis in response to UVB, fail to upregulate Puma and Noxa to induce apoptosis but can enter senescence by the upregulation of p21^Waf1/Cip1^ [28]. However, the upregulation of Puma has been observed in senescent IMR90 cells [29] and on entry into senescence after Cdk2ap1 knockdown [60] (Figure 3B). Bim shows low expression in aged peripheral naive CD4 T cells exhibiting higher levels of p16^INK4a^ and p19^ARF^ [66], as well as in senescence in K562 cells [31]. Additionally, Bim expression is reduced in spindle mitotic stress induced by deletion of the transforming centrosomal acidic coiled-coil protein (TACC)3, which links microtubule integrity to spindle poison-induced cell death, G1 cell arrest and the upregulation of nuclear p21^Waf1/Cip1^ [67]. However, the upregulation of Bim has been observed in senescent IMR90 cells [29], as well as after the treatment of uveal melanoma with a combination of mitogen-activated protein kinase kinase inhibitors (MEKi) with a DNA methyltransferase inhibitor (DNMTi) that induces an increase in p21^Waf1/Cip1^ expression [68] (Figure 3B). The relative Bad levels were elevated from 60% to 130% in prolonged senescent cultures of porcine pulmonary artery endothelial cells (PAECs), whereas the steady-state Bcl-2 levels decreased to less than 20% favoring cell death [69]. On the other hand, Bad influences carcinogenesis and cancer chemoresistance. When unphosphorylated, Bad dimerizes with Bcl-xL and Bcl-2, releasing Bax and allowing the ignition of apoptosis. However, when phosphorylated, Bad (pBad) is unable to heterodimerize with Bcl-2 or Bcl-xL, and therefore, Bax is not released to initiate apoptosis. Higher levels of pBad have been observed in normal immortalized cells compared with tumor cells [70]. Bmf is a functional target of miR-34c-5p, and long noncoding RNA (lncRNA)-ES3 acts as a competing endogenous RNA (ceRNA) of miR-34c-5p to regulate the expression of Bmf in human aorta VSMCs. Thus, the lncRNA-ES3/miR-34c-5p/Bmf axis upregulates calcification/senescence of vascular smooth muscle cells (VSMCs) [71] (Figure 3B). Bnip3 is activated in hypoxic human papillomavirus type 16 (HPV16)-positive cervical cancer cells, allowing the evasion of senescence [72]. Overexpressing Bnip3 fibroblasts show the key features of a senescence phenotype, such as the induction of p21^Waf1/Cip1^ and p16^Ink4a^, cell hypertrophy and the upregulation of β-galactosidase activity [73] (Figure 3B). Urolithin A attenuates auditory cell senescence by activating mitophagy. However, the ablation of Bnip3, which acts as a mitophagy-related gene, results in the abrogation of UA-induced anti-senescent activity [74].

## 4. Senescence and Bcl-2 Family Proteins in Tetraploid Cells

Although the state of tetraploidy, two complete sets of chromosomes, is frequent in the development and differentiation of specialized cell types (i.e., hepatocytes), almost 100 years ago, Theodor Boveri proposed that tetraploid cells are the precursors of aneuploid cancer cells [75]. Cytokinesis failure, cell fusion, mitotic slippage, endoreplication or cell cannibalism are mechanisms that can give rise to tetraploid/polyploid cells [76,77]. Normally, tetraploid cells enter proliferative arrest or apoptosis due to the p53- and pRb-dependent tetraploidy checkpoints [78]. Failure to arrest at a tetraploidy checkpoint can facilitate the appearance of aneuploid cells due to chromosome loss, asymmetric cell divisions and/or multipolar mitoses [76]. Induced tetraploid primary cells can enter senescence independent of the DNA damage pathway but are dependent on p16^INK4a^ expression. Indeed, the suppression of p16^INK4a^ prevents cycle arrest when tetraploidy is induced [79]. Alternatively, malignant tetraploid cells may arise driven by persistent telomere dysfunction. According to this model, at the onset of tumorigenesis, cells with telomere shortening may undergo a DNA damage signal that causes permanent cell cycle arrest followed by entry into senescence or activation of the apoptotic cascade. Tetraploid cells would emerge by escaping the cycle arrest and bypassing mitosis to re-enter into the S phase [3]. Although the correlation between polyploidy and resistance to Bcl-xL-mediated apoptosis has not been reported, it has been described that the inhibition of Bcl-xL reduces the viability of polyploid lymph node carcinoma of the prostate cancer cells and that the survival of polyploid tumor cells depends on Bcl-xL [80] (Figure 4A). Our group has described the paradoxical implication of Bax and Bak in the persistence of tetraploid mouse embryonic fibroblasts (MEFs). The absence of Bax and Bak limits the protection against microtubule inhibitors. Furthermore, although both wild-type and Bax/Bak (DKO) MEFs accumulate tetraploid cells in a similar manner, purified tetraploid DKO MEFs fail to resume proliferation. No such effects were observed in the MEFs deficient for Bax or Bak individually; Noxa; Bim; Puma or combinations of Puma/Noxa, Bim/Puma or Bim/Bid. Tetraploid Bax/Bak DKO MEF fail to proliferate, because they enter senescence. Indeed, tetraploid DKO MEFs show increased β-galactosidase activity and higher levels of p16 ^Ink4a^, p21^Waf1/Cip1^ and p27 ^Kip1^ (Figure 4B). Interestingly, the introduction of Bak into the endoplasmic reticulum reduces the accumulation of senescence markers levels, such as p21^Waf1/Cip1^ and p27 ^Kip1^, and restores the proliferative capacity of tetraploid DKO MEFs [12]. Therefore, Bak plays a determinant role in tetraploidy-induced senescence.

In multiple human cancers (adenocarcinoma of liver, pancreas, lung, prostate, colon, ovary, esophagus and breast, as well as cervical and bladder carcinoma), evidence of tetraploidization, loss of p53/pRb and telomerase activation have been observed [81]. The suppression of p53, p21^Waf1/Cip1^ or Bax, as well as the overexpression of Bcl-2, favor the appearance and survival of tetraploid cells (Figure 4A) [82]. In addition, the immune system controls cancer cell ploidy. Tetraploid tumor cells exhibit increased exposure of the calcium chaperone calreticulin (CALR) to the plasma membrane, where it acts as an “eat me” signal to the immune system. CALR translocation from the endoplasmic reticulum to the plasma membrane is mediated, among others, by the activation of Bax and Bak [83,84]. Tetraploid (or near-tetraploid) cells that manage to escape the control exercised by the immune system show an immunoselection characterized by a loss in the DNA content and a lower expression of CALR at the cell surface [83,85]. Conversely, human colon cancer cells HCT116 genetically modified to suppress Bax and Bak (HCT116 DKO cells) exhibit a reduced proliferative capacity and induced entry into senescence after tetraploidization [12].

The emergence of polyploid cancer cells can be triggered using different stressors, such as alkylating agents, platinum-based drugs, antimetabolites, topoisomerase inhibitors, microtubule inhibitors, mTOR inhibitors, poly (ADP-ribose) polymerase inhibitors, radiation, hypoxia or ROS modulators [77]. Certain chemotherapeutic agents, such as microtubule inhibitors, are particularly prone to cause the generation of tetraploid cells. Taxanes (such as paclitaxel and docetaxel) stabilize microtubules, resulting in multipolar spindles, whereas vinca alkaloids (such as vinblastine and vincristine) inhibit microtubule assembly [6] (Figure 4C). Interestingly, paclitaxel can induce the cell cycle arrest of cancer cells, while senescence may increase the resistance to paclitaxel [86]. Antiapoptotic members of the Bcl-2 family are often amplified during carcinogenesis, which can lead to a resistance to microtubule inhibitors. Nevertheless, the use of microtubule inhibitors increases the post-translational modification of the antiapoptotic Bcl-2 family. Nonetheless, differences have been observed between solid tumors and hematopoietic tumors. In solid tumors, elevated Bcl-2 protein expression has been shown to increase the sensitivity to microtubule inhibitors, probably due to increased expression of the proapoptotic BH3 protein, Bim. Thus, a loss of Bcl-2 confers resistance to microtubule inhibitors. The opposite effect is observed in hematopoietic tumors, in which Bcl-2 overexpression protects them from microtubule inhibitor-induced apoptosis, whereas treatment with microtubule inhibitors decreases Bcl-2 expression. On the other hand, the combination of Bcl-xL inhibitors with taxanes produces a synergistic response. The deletion of Bcl-w increases the rate of paclitaxel-induced cell death, whereas the overexpression of Bcl-w promotes paclitaxel resistance. Finally, Mcl-1 has been closely linked to resistance to microtubule inhibitors. When Mcl-1 protein levels are low or absent, apoptosis is favored. However, when Mcl-1 is overexpressed, it favors cell viability [55] (Figure 4C).

Tetraploid cancer cells are relatively resistant to radio- and chemotherapy [82]. One possibility is that tetraploid cells escape cytotoxicity by stopping their proliferation and re-entering the cell cycle with a chemoresistant phenotype. A prolonged senescent period would allow the cells to achieve the necessary adaptation, through transcription and translation of the necessary proteins, to their new ploidy state [87]. Multiple lines of evidence show that a subpopulation of senescent tumor cells induced by therapies, usually DNA-damaging such as camptothecin, doxorubicin or cisplatin, often develop transient polyploidy. This polyploidy may contribute, in part, to the ability of tumor cells to surmount senescent growth arrest [80]. Thus, it is known that mitotic spindle stress triggers paclitaxel sensitivity by entering into premature senescence [67]. The combined treatment of vinblastine with interferon (IFN)-α increases p21^Waf1/Cip1^ expression and Bak levels in human melanoma (M14) cells [88]. TIS can promote the formation of polyploid senescent cells associated with reduced cyclin-dependent kinase (Cdk) 1 expression, which is modulated by p21^Waf1/Cip1^ and p27^Kip1^. While p21^Waf1/Cip1^ inhibits apoptosis, p27^Kip1^ prevents the formation of polyploid cells in TIS [89] (Figure 4C).

## 5. The Bcl-2 Family Proteins as a Target for Senolytic Agents

TIS has been identified after radiation or genotoxic chemotherapy. Arrest can occur in both G1 and G2/M and is characterized by an increased expression of p16^Ink4a^, p21^Waf1/Cip1^ and p27^Kip1^. TIS may function as an alternative onco-suppressive mechanism when apoptotic pathways are disabled. Moreover, TIS can induce persistent cell cycle arrest at any stage of tumor development [90]. However, the data obtained in recent years show the danger of senescent cancer cells, since (1) senescent cells can escape growth arrest and resume cell proliferation, (2) senescent cancer cells that manage to escape arrest exhibit stem cell-like characteristics and (3) senescent tumor cells may escape recognition and elimination by the immune system [80].

Senolytics are senotherapeutics that selectively eliminate senescent cells [91,92,93]. Inhibitors of the Bcl-2 family have been identified among the different types of compounds with senolytic activity [13] (Table 1). ABT-199 (Venetoclax) targets only the Bcl-2 protein [94]. ABT-263 (also known as Navitoclax) is an orally bioavailable Bad-like BH3 mimetic. ABT-263 maintains a high affinity for Bcl-2, Bcl-w and, especially, Bcl-xL. Reportedly, ABT-263 inhibits the interaction of Bcl-2 and Bcl-xL, leading to the release of Bim, as well as to trigger the translocation of Bax, initiating the intrinsic pathway of apoptosis [95]. The addition of senolytic agents such as ABT-199 or ABT-263 after irradiation induces apoptotic cell death in soft tissue sarcomas (STS), which undergo TIS with increased levels of the antiapoptotic Bcl-2 family [96]. The combination of gemcitabine with everolimus or ionizing radiation induces the senescence of malignant meningioma cells, which are eliminated with ABT-263 [97,98]. Wogonin, a well-known natural flavonoid compound, induces cellular senescence in T-cell malignancies and activates DDR mediated by p53, as well as the upregulated expression of Bcl-2 in senescent T cells. ABT-263 induces apoptotic cell death in wogonin-induced senescent cells [99]. In 2020, Muenchow et al. proposed a combinatorial treatment of A-199 and the proteosome inhibitor bortezomib (BZB) against STS, resulting in a sensitization to apoptosis by the simultaneous release of proapoptotic proteins such as Bax, Bok and Noxa and inhibition of Mcl-1 [100]. ABT-263 is an effective senolytic in senescent human umbilical vein epithelial cells (HUVECs) and IMR90 cells [101], irradiated or old normal murine senescent bone marrow hematopoietic stem cells and senescent muscle stem cells [102] and prostate cancer TIS [9]. The sequential combination of TIS and ABT-263 redirects the response towards apoptosis by interfering with the interaction between Bcl-xL and Bax [103]. Breast cancer *Tp53^+/+^* cells depend on Bcl-xL to survive TIS. These cells can be killed using ABT-263, although sensitivity takes days to develop. However, a low expression of Noxa confers resistance to ABT-263 in some cells, requiring the additional inhibition of Mcl-1 [104].

ABT-737, a small molecule cell-permeable Bcl-2 antagonist that acts as a BH3 mimetic, inhibits Bcl-2, Bcl-w and Bcl-xL proteins, causing the preferential apoptosis of senescent cells induced by DNA damage [24,29]. Although the mechanism of action of ABT-737 has not been described in detail, it is known that ABT-737 inhibits the protective effect of Bcl-2 and Bcl-xL, an effect that is dependent on Bax or Bak, and activates the cleavage of caspases 8/9 in multiple myeloma cells [105]. ABT-737 eliminates Cox2-expressing senescent cells from PanIn lesions [106], and both ABT-737 and Navitoclax have shown a senolytic effect on senescent glioblastoma cells induced by the DNA-methylating drug temozolomide (TMZ) [107].

A1331852, a small molecule BH3 mimetic, inhibits Bcl-xL. Radiation plus TMZ is a common treatment in GBM that induces a state of senescence and sustained proliferative arrest. The use of Bcl-xL inhibitors (A1331852, A1155463 and A-263) increases the vulnerability of GBM to TMZ treatment [50]. However, the use of ABT-199 plus TMZ has shown contradictory effects in GBM [50,108]. Since Bcl-xL has been observed to be upregulated in senescent cholangiocytes induced by ionizing radiation, A1331852 reduces its presence by 80% [109], whereas Bak plays a key role in A-1331852-induced apoptosis in senescent chondrocytes [110]. The treatment of Dox combined with A-1331852 in different subcutaneous xenograft models of solid tumors shows the disruption of Bcl-xL:Bim complexes and induces cytochrome *c* release, activation of caspases 3/7 and externalization of phosphatidylserine, features of apoptosis [111,112]. A-1331852 upregulates the expression of Bid and Bax. In fact, A-1331852 promotes the apoptosis of senescent human lung A549 cells by influencing the interaction between Bcl-xL and Bid and that between Bcl-xL and Bax [23]. Both A1331852 and A1155463 are senolytic for ionizing radiation-induced senescent HUVECs and IMR90 cells. Treatment with A1155463 after ionizing irradiation also induces the cleavage of caspase 3/7 [113].

Other types of Bcl-2 family inhibitors that act as senolytic agents are small molecule Mcl-1 inhibitors such as A1210477 and S63845. A1210477 synergizes with EE-84, an aplysinopsin that induces a senescent phenotype in K562 cells [114]. Recently, it has been described that treatment with the Mcl-1 inhibitor S63845 leads to the complete elimination of senescent tumor cells and metastases [53]. The treatment of myeloma with A1210477 has been shown to disrupt Mcl-1/Bak complexes, and Bak release would promote cell death. However, free Bak can be recaptured by Bcl-xL, leading to a resistance to A1210477 [115]. Similarly, S63845-induced apoptosis occurs in a Bak-dependent manner in solid tumor-derived cell lines [116]. S64315 enhances the selective senolytic effect of ABT-263 and ABT-737. Radiation-induced retinal pigment epithelium senescent cells that survive treatment with the selective Mcl-1 inhibitor have been found to express increased levels of the Bcl-xL protein [13]. The combination of inhibitors of antiapoptotic proteins of the Bcl-2 family with taxanes and vinca alkaloids increases the efficacy of microtubule inhibitors, which would make it possible to reduce the doses of these chemotherapeutic agents while reducing their toxicity [55].

According to Wei et al., the antiapoptotic proteins Bcl-2, Bcl-xL and Mcl-1 are bound to the multidomain proapoptotic proteins Bax and Bak, inhibiting their activation. Following a cellular stress stimulus, the expression of proapoptotic BH3-only proteins Bad and Noxa is increased. Bad binds preferentially to Bcl-2 and Bcl-xL, whereas Noxa binds preferentially to Mcl-1. In consequence, Bad and Noxa free Bax and Bak from binding to antiapoptotic proteins and activating them, thus initiating the apoptotic pathway [117]. Since antiapoptotic Bcl-2 family proteins are upregulated in irradiation-induced senescent cells, it is pertinent to propose combination treatments with Bcl-2 family inhibitors acting as senolytic agents to achieve effective Bax and Bak release and senescent cell death.

Senolytic agents other than Bcl-2 family inhibitors may also involve Bcl-2 family proteins in their mechanisms of action. Nintedanib, a tyrosine kinase inhibitor, induces apoptosis in triple-negative breast cancer cells [118], inhibits tumor growth of malignant pleural mesothelioma [119] and non-small cell lung cancer [120] and is one of two US Food and Drug Administration-approved treatments for idiopathic pulmonary fibrosis [121]. The senolytic effect of Nintedanib, which induces Bim expression, as well as the cleavage of caspase 9 and downstream factors caspases 3/7 prominently in senescent cells compared to non-senescent cells, is mediated by signal transducer and activator of transcription 3 (Stat3) inhibition [122]. The inhibition of ubiquitin-specific peptidase 7 selectively induces the apoptosis of senescent cells. The mechanisms of action include the ubiquitination and degradation of the human homolog of Mdm2 and the consequent increase in p53 levels, which, in turn, induces the proapoptotic proteins Puma and Noxa, among others, and inhibits the interaction of Bcl-xL and Bak, selectively inducing apoptosis in senescent cells [123].

**Table 1 ijms-24-06374-t001:** Senolytic agents targeting Bcl-2 family proteins alone or in combinatorial therapy and their modes of action in cancer treatments.

Therapy + Senolytic Agent + Other Agents	Disease	Target/s	Mode of Action	References
ABT-199		Bcl-2	Releases Bax from Bcl-2 inhibition	
ABT-263		Bcl-2	Inhibition of Bcl-2-Bcl-xL interaction	
		Bcl-w	Release of Bim	
		Bcl-xL	Bax translocation	
			Apoptosis	
Irradiation + ABT-199 or ABT-263	Soft tissue sarcomas		Therapy-Induced Senescence	[93]
			Apoptosis	
Gemcitabine + Everolimus or Ionizing radiation + ABT-263	Malignant meningioma		Senescence	[94,95]
			Apoptosis	
Wogonin + ABT-263	T-cell malignancies		p53-mediated DNA damage	[96]
			Upregulation of Bcl-2	
			Apoptosis	
ABT-199 + Bortezomib	Soft tissue sarcomas		Release of Bax from Bcl-2 inhibition	[97]
			Accumulation of Bok and Noxa	
			Inhibition of Mcl-1	
			Apoptosis	
ABT-263	Senescent HUVECs		Senolytic effect	[98]
	IMR90 cells			
	Senescent MuSCs			
DNA damage inducers + ABT-263	Prostate cancer-TIS		Interferes with the interaction between Bcl-xL and Bax	[100]
			Apoptosis	
ABT-263	Breast cancer *Tp53+/+*			[101]
ABT-737		Bcl-2	Inhibits the protective effect of Bcl-2 and Bcl-xL	[23,28,102]
		Bcl-w	activate the cleavage of caspases 8/9	
		Bcl-xL	Mechanism of action has not been described in detail	
ABT-737	PanIn lesions		Eliminates Cox2-expressing senescent cells	[103]
TMZ + ABT-737 or ABT-263	Senescent glioblastoma cells		Senolytic effect	[104]
Radiation+ TMZ + Bcl-xL inhibitors (ABT-263, A1331852, A1155463)	GBM		Increases the vulnerability of GBM to TMZ treatment	[43]
TMZ + ABT-199	GBM		Senolytic effect	[105]
Dox + A1331852	Xenograft models of solid tumors		Disruption of Bcl-xL:Bim complexes	[108,109]
			Induces cytochrome c release	
			Activation caspases 3/7	
			Externalization of phosphatidylserine	
			Apoptosis	
A1331852		Bid	Upregulate the expression of Bid and Bax	[22]
		Bax		
Ionizing radiation + A1331852 or A1155463	HUVECs		Induces cleavage of caspase 3/7	[110]
	IMR90 cells			
A1210477		Mcl-1	Mcl-1 inhibitors	[111]
S63845				
A1210477 + EE-84	Chronic myelogenous leukemia		Synergistic effect	[111]
S63845	Senescent tumor cells and metastases		Complete elimination	[50]
A1210477	Myeloma		Disrupt Mcl-1/Bak complexes	[112]
			Apoptosis	
Taxanes + Vinca alkaloids + Combination of antiapoptotic proteins of the BCL-2 family inhibitors			Increases the efficacy of microtubule inhibitors	[52]
Nintedanib	Triple-negative breast cancer cells	Tyrosine kinase inhibitor	Induces apoptosis	[115,116,117,119]
	Malignant pleural mesothelioma		Inhibits tumor growth	
	Non-small cell lung cancer		Bim expression	
			Cleavage of Caspase-9 and caspases 3/7	
P5091		USP7 inhibitor	Promotes ubiquitination and degradation of Mdm2	[120]
			Increases p53, Puma and Noxa	
			Inhibits the interaction of Bcl-xL and Bak	
			Apoptosis in senescent cells	

HUVECs, human umbilical vein epithelial cells; MuSCs, muscular stem cells; TIS, therapy-induced senescence; PanIn, pancreatic intraepithelial neoplasia; TMZ, temozolomide; GBM, glioblastoma multiforme; Dox, doxorubicin.

## 6. Conclusions and Future Perspectives

Bcl-2 family proteins affect the regulation of cellular senescence. The relationship of Bcl-2 family proteins to senescence is dual. Thus, the expression of these proteins influences their entry into senescence, but the induction to senescence simultaneously alters the expression of Bcl-2 family proteins. In general, although with some exception, entry into senescence coincides with the upregulation of antiapoptotic proteins, especially Bcl-2, Bcl-xL and Mcl-1, as well as with the downregulation of proapoptotic proteins. This explains why senescent cells are relatively resistant to cell death. Targeting Bcl-2, Bcl-xL, Mcl-1 and, to a lesser extent, Bcl-w can be used for the selective elimination of senescent cells or senolysis.

In addition, proteins of the Bcl-2 protein family regulate the formation/persistence of tetraploid cells. The overexpression of Bcl-2 and under-expression of Bax favor the appearance of tetraploid cells. However, the combined absence of Bax and Bak limits the proliferation of tetraploid cells due to their entry into senescence. Importantly, the restoration of Bak in the endoplasmic reticulum is sufficient to avoid senescence and, hence, to enhance the proliferative capacity of tetraploid cells. Therefore, it seems that Bak plays a key role in the progression of tetraploid cells towards a malignant state. Whilst it is true that mice deficient in both Bax and Bak do not develop a malignant disease as a primary phenotype, it would be interesting to explore the possible involvement of Bak in the development of cancer. In view of the accumulated data, it is tempting to speculate that a cancer that is deficient for Bak, and therefore resistant to apoptosis, may be treated with a combination of antimitotic agents plus a senolytic agent for its elimination.

Ultimately, this review may facilitate the interpretation of bioinformatics data, as well as highlight the interest in further analyses aimed at opening up new therapeutic possibilities.

## Figures and Tables

**Figure 1 ijms-24-06374-f001:**
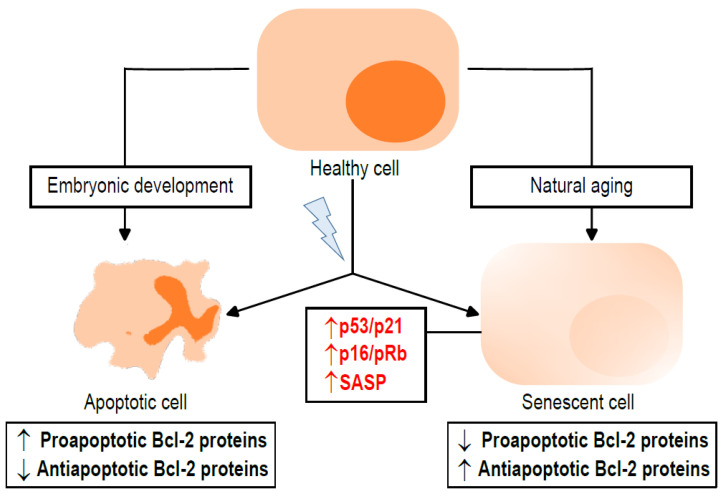
Apoptosis and senescence: two ways to suppress cell proliferation. Under physiological conditions, such as embryonic development or natural aging, healthy cells can enter apoptosis or senescence, respectively. However, apoptosis and senescence also play a role in pathophysiological situations. Certain stimuli can provoke stress signals, such as DNA damage or increased reactive oxygen species (ROS), leading to the suppression of cell proliferation by apoptosis or senescence. B-cell lymphoma 2 (Bcl-2) family proteins are differentially expressed in each case. Apoptosis is characterized by the increased expression of proapoptotic proteins and decreased expression of antiapoptotic proteins, whereas, in senescence, there is generally an increase in antiapoptotic proteins and a decrease in proapoptotic proteins. Likewise, senescent cells show an increase in senescent markers, such as activation of the p53/p21^Waf/Cip1^ (p53/p21), p16^INK4a^/retinoblastoma protein (p16/pRb) and senescence-associated secretory phenotype (SASP) pathways. Up arrows mean increase, down arrows mean decrease. Black arrows are related to Bcl-2 family proteins, while red arrows are related to senescence hallmarks.

**Figure 2 ijms-24-06374-f002:**
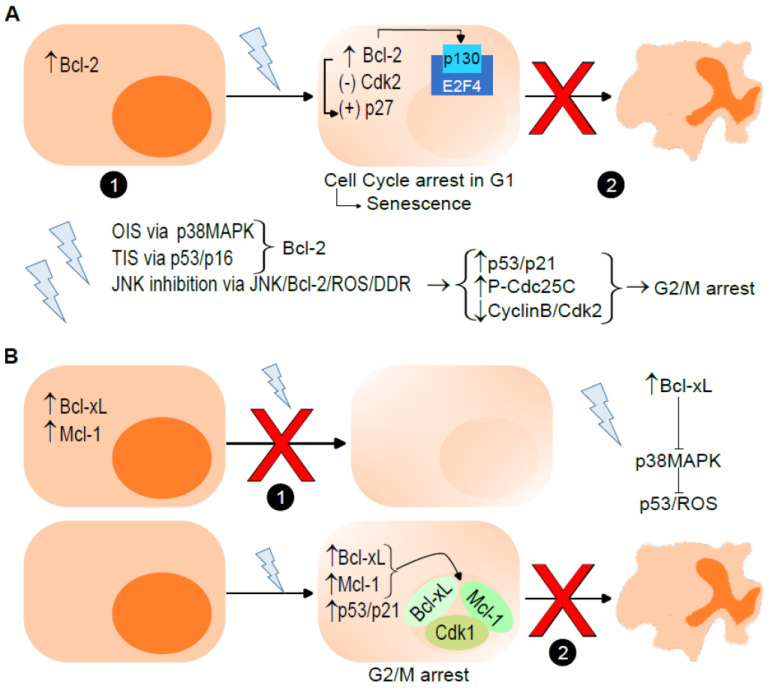
Role of the antiapoptotic proteins of the Bcl-2 family. (**A**) High levels of Bcl-2 trigger senescence in response to several stimuli (1), while Bcl-2 overexpression confers the resistance of senescent cells to apoptosis (2). Overexpression of Bcl-2, inhibition of cyclin-dependent kinase (Cdk)2 and induction of p27^Kip1^ (p27) lead to cell cycle arrest in G1. Moreover, Bcl-2 upregulates p27 ^Kip1^ and p130, which forms repressive complexes with transcriptor factor E2F4, inhibiting its release and preventing cell cycle progression. Oncogene-induced senescence (OIS) and therapy-induced senescence (TIS) increase Bcl-2 expression via p38 mitogen-activated protein kinase (p38MAPK) and via p53/p16^INK4a^ (p53/p16), respectively. Jun N-terminal kinase (JNK) inhibition leads to the dephosphorylation of Bcl-2, accumulation of ROS, induction of DNA damage response (DDR) and G2/M arrest, which is characterized by an increase in the p53/p21 pathway, inactivation of M-phase inducer phosphatase 3 (Cdc25C, P-Cdc25C) and a reduction of cyclin B/Cdk2. ROS, reactive oxygen species; DDR, DNA damage response. (**B**) Overexpression of B-cell lymphoma extra-large (Bcl-xL) or Induced myeloid leukemia cell differentiation protein (Mcl-1) reduces entry into senescence (1). A possibility is that Bcl-xL blocks p38MAPK activation and inhibits senescence induction by preventing p53-induced ROS generation. Treatment with G2/M blocking agents causes translocation of Bcl-xL and/or Mcl-1 to the nucleus, where they bind to Cdk1-stabilizing senescence. Senescent cells showing elevated Bcl-xL or Mcl-1 levels prevent apoptosis (2). Up arrows mean increase, down arrows mean decrease.

**Figure 3 ijms-24-06374-f003:**
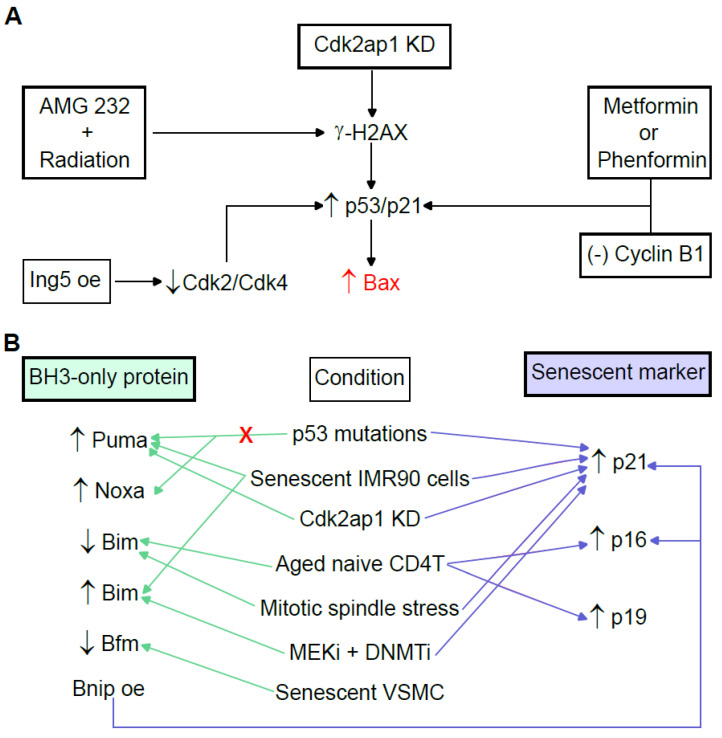
Role of the proapoptotic proteins of the Bcl-2 family. (**A**) Different stimuli activate the H2A histone family member X (γ-H2AX)/p53/p21/Bax pathway at different levels. The Cdk2ap1 knockdown (Cdk2ap1 KD) and combinatorial treatment of AMG-232 with radiation causes an increase in γ-H2AX. Treatment with metformin or phenformin, as well as inhibition of cyclin B1, causes an increase in the expression of p53 and/or p21. Overexpression of Ing5 (Ing5 oe) results in a lower expression of Cdk2/Cdk4 and higher expression of p53/p21. As a result, senescent cells with elevated Bax expression are obtained in all cases. (**B**) Different senescence conditions result in the differentially modified expression of proapoptotic BH3-only proteins, as well as increased expression of senescent markers. Cdk2ap1, Cdk2-associated protein-1; Ing5, inhibitor of growth protein 5; MEKi, mitogen-activated protein kinase kinase inhibitors; DNMTi, DNA methyltransferase inhibitor; VSMC, vascular smooth muscle cells. Up arrows mean increase, down arrows mean decrease.

**Figure 4 ijms-24-06374-f004:**
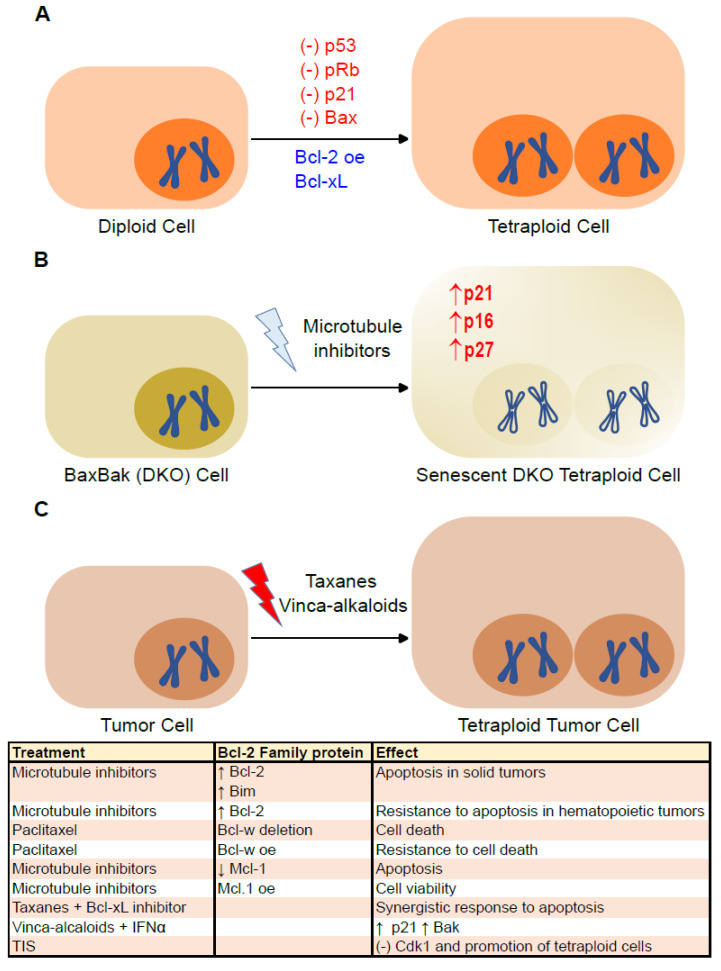
Involvement of Bcl-2 family proteins and senescence in the emergence of tetraploid cells. (**A**) The generation of tetraploid cells is favored by the absence of p53, retinoblastoma protein (pRb), p21^Waf1/Cip1^ (p21) and Bax, as well as the overexpression of Bcl-2 and the presence of Bcl-xL. (**B**) Cells deficient in Bax and Bak enter senescence when tetraploidized with microtubule inhibitors and show an increase in p21^Waf1/Cip1^, p16^INK4a^ (p16) and p27^Kip1^ (p27). (**C**) Some chemotherapies, such as taxanes and vinca alkaloids, induce tumor cell polyploidization and cell cycle arrest. The table summarizes the relationship found between tetra- or polyploidy-inducing chemotherapies, Bcl-2 family proteins and their effects on tumor cells. Up arrows mean increase, down arrows mean decrease.

## Data Availability

Data sharing not applicable.

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
