# Peer review of "Involvement of Bcl-2 Family Proteins in Tetraploidization-Related Senescence"

_ijms, 2023, doi:10.3390/ijms24076374_

Round 1

Reviewer 1 Report

In general, the review is well written. However, each paragraph is full of detailed info and sometimes it can be confusing and repetitive. It lacks a general point of view on the process that is described, it seems more just a description of papers.

Minor comments:

Figure 1 is missing

check citation typos in the text: e.g cit 8 with PMID

page 9:correct b-galactose

Regarding the section "3. Role of the Bcl-2 Family in Senescence" and especially 3.3. BH3-only proapoptotic Bcl-2 proteins" I think is a bit confusing and it is difficult to understand the point of the paragraph, try to be more clear

Improve quality of fig 4 - figure and table

Reviewer 2 Report

In this Review the Authors focused on the role of BCl proteins in senescence. Using extensive analysis of the literature, they carefully described mutual interactions between entry into senesce entry and Bcl proteins. They also provided new evidence about contribution of BCL proteins in regulation of apoptosis, survival and senescence in general and in tetraploid cells.

This is a comprehensive and well written Review. It is easy for perception despite the description of many complicated molecular interactions. It also provides a lot of new and evidence of great important for clinical medicine.  For example, it was very interesting to find out that senescence is associated with increased apoptosis resistance and that senescent tumor cells may escape recognition and elimination by the immune system.  The evidence provided in the Review will help in the interpretation of bioinformatics data.

This review can be published.

There are only several small minor points.

Abstract.

1.      Please, change a style little bit from narrative to more scientific.

2.      To do this, please outline the aim of the Review after describing background.

3.      Please, underline brightest analytical openings and their novelty and importance in therapy.

Results  

1 Please, insert figure 1

2 Please, provide missing references for the second paragraph of the chapter  “2. Apoptosis and Senescence: Two Ways to Suppress Cell Proliferation” and for the first and second paragraphs of the chapter “3. Role of the Bcl-2 Family in Senescence”
